# Seroprevalence of SARS-CoV-2 antibodies and retrospective mortality in a refugee camp, Dagahaley, Kenya

Etienne Gignoux[1]*, Frida Athanassiadis[2], Ahmed Garat Yarrow[2], Abdullahi Jimale[2], Nicole Mubuto[2], Carole Déglise[2], Denis Onsongo Mosoti[3], Andrew S. Azman[2,4,5], Matilu Mwau[6], Francisco Luquero[1], Iza Ciglenecki[2]

1 Epicentre, Paris, France, 2 Médecins Sans Frontières, Geneva, Switzerland, 3 Ministry of Health, Nairobi, Republic of Kenya, 4 Department of Epidemiology, Johns Hopkins Bloomberg School of Public Health, Baltimore, Maryland, United States of America, 5 Institute for Global Health, Faculty of Medicine, University of Geneva, Geneva, Switzerland, 6 Kenya Medical Research Institute, Nairobi, Kenya

* Etienne.GIGNOUX@geneva.msf.org

## Abstract

### Background

Camps of forcibly displaced populations are considered to be at risk of large COVID-19 outbreaks. Low screening rates and limited surveillance led us to conduct a study in Dagahaley camp, located in the Dadaab refugee complex in Kenya to estimate SARS-COV-2 seroprevalence and, mortality and to identify changes in access to care during the pandemic.

### Methods

To estimate seroprevalence, a cross-sectional survey was conducted among a sample of individuals (n = 587) seeking care at the two main health centres and among all household members (n = 619) of community health workers and traditional birth attendants working in the camp. A rapid immunologic assay was used (BIOSYNEX® COVID-19 BSS [IgG/IgM]) and adjusted for test performance and mismatch between the sampled population and that of the general camp population. To estimate mortality, all households (n = 12860) were exhaustively interviewed in the camp about deaths occurring from January 2019 through March 2021.

### Results

In total 1206 participants were included in the seroprevalence study, 8% (95% CI: 6.6%-9.7%) had a positive serologic test. After adjusting for test performance and standardizing on age, a seroprevalence of 5.8% was estimated (95% CI: 1.6%-8.4%). The mortality rate for 10,000 persons per day was 0.05 (95% CI 0.05–0.06) prior to the pandemic and 0.07 (95% CI 0.06–0.08) during the pandemic, representing a significant 42% increase (p<0.001). Médecins Sans Frontières health centre consultations and hospital admissions decreased by 38% and 37% respectively.

**Data Availability Statement:** The minimal dataset underlying the findings of this study is available on request, in accordance with the legal framework set forth by Médecins Sans Frontières (MSF) data

sharing policy (Karunakara U, PLoS Med 2013). MSF is committed to share and disseminate health data from its programs and research in an open, timely, and transparent manner in order to promote health benefits for populations while respecting ethical and legal obligations towards patients, research participants, and their communities. The MSF data sharing policy ensures that data will be available upon request to interested researchers while addressing all security, legal, and ethical concerns. All readers may contact the generic address data. sharing@msf.org or Ms. Aminata Ndiaye (aminata. ndiaye@epicentre.msf.org) to request data.

**Funding:** The study was funded by the Operational Centre of Geneva of Médecins Sans Frontières. The funders had no role in study design, data collection, and analysis, decision to publish, or preparation of the manuscript.

**Competing interests:** The authors have declared that no competing interests exist.

**Abbreviations:** CHW, Community health worker; CHWs, Community health workers; CI, Confidence interval; CMR, Crude mortality rate; IgG, Immunoglobulin G; IgM, Immunoglobulin M; MPH, Ministry of Public Health; MSF, Médecins Sans Frontières; RR, Relative Risk; TBA, Traditional birth attendant; WHO, World Health Organization; PCR, Polymerase Chain Reaction; ELISA, Enzyme-linked Immunosorbent Assay.

## Conclusion

The number of infected people was estimated 67 times higher than the number of reported cases. Participants aged 50 years or more were among the most affected. The mortality survey shows an increase in the mortality rate during the pandemic compared to before the pandemic. A decline in attendance at health facilities was observed and sustained despite the easing of restrictions.

## Background

Many predicted that resource-limited settings would be particularly hard hit by the COVID-19 epidemic given the difficulty of imposing confinement measures as well as the reduced access to diagnostics and health care in these contexts [1]. Overcrowded places such as urban slums and camps of forcibly displaced populations were of particular concern. In these settings, in addition to population density, higher transmissibility could occur due to larger household sizes, intense social mixing between the young and elderly, inadequate water and sanitation, and specific cultural and faith practices [2].

This has not materialized in most camps of forcibly displaced people, where the number of reported cases and deaths has been much lower than feared [1]. The low numbers have been attributed to limited testing capacity, differences in the population structure with a small proportion of elderly at high risk of severe disease and death, a predominance of asymptomatic and pauci-symptomatic infections, early implementation of confinement [3], social structure leading to different epidemic dynamics [4], or other unknown factors associated with this population and the context. However, the actual impact of COVID-19 on these populations remains an open question.

Therefore, the aim of the study was to provide a more accurate picture of the extent of the epidemic, the specific objectives were to estimate the seroprevalence of SARS-COV-2 through a cross-sectional survey; in addition, through a retrospective survey and programmatic data, to assess its impact on mortality and access to care before and during the COVID-19 epidemic. The Dagahaley camp is part of the Dadaab refugee complex in Kenya, where Médecins Sans Frontières (MSF) has been working since 2009, was a relevant location for the study. MSF provides in-patient and out-patient health services in the hospital and two health centres located in the camp and works with a network of 110 community health workers (CHW) and 45 traditional birth attendants (TBA). The population of Dagahaley camp was estimated at 72,635 inhabitants, during a survey conducted in September 2018 (personal communication Etienne Gignoux).

As of the start of the study on 3 March 2021, 106,470 confirmed cases of COVID-19 had been reported in Kenya including 1,863 deaths (source World Health Organization -WHO). Based on the 2019 national census, this corresponds to an attack rate of 0.22% and 3.9 COVID-19 related deaths per 100,000 population. The first confirmed case of COVID-19 in Dagahaley camp was identified on 16 May 2020. Few cases were reported in Dagahaley camp in May and June 2020; then a first peak occurred in September and a second increase started in February 2021. As of 3 March 2021, 940 Polymerase Chain Reaction (PCR) tests had been performed in Dagahaley camp, confirming 47 positive cases, including 3 deaths.

## Methods

### Design and setting

The study entailed a cross-sectional survey using convenience sampling to estimate seroprevalence. A population based survey was not deemed feasible in this setting due to limited access

by MSF staff to the camp and expectedly low participation rates in the camp. To estimate sero-prevalence, two cohorts accessible to trained MSF staff were included: a) All CHWs and TBAs working in Dagahaley camp and their household members, as well as b) Patients and caretakers presenting at MSF-supported health centres in Dagahaley camp.

In order to introduce the study and seek participants, trained study staff organized a meeting with all CHWs and TBAs collaborating with MSF. They highlighted the volunteer nature of participation in the study. When he/she agreed, he/she informed his/her family members of the possibility to participate in the study and gave them an appointment time to present to MSF health facilities. For patients and caretakers, a systematic time sampling was conducted of patients and caretakers presenting to health centres stratified by age.

For practical reasons in a context where access is limited, a rapid antibody test was chosen. Antibody test haven't shown important discordances with the Enzyme-linked Immunosorbent Assay (ELISA), based on the correlations observed in the published literature (3). The test used in this study (BIOSYNEX® COVID-19 BSS [IgG/IgM]) offers good performance (sensitivity: 95.8% (95% CI: 90.2–100.0), specificity: 98.1% (95% CI: 94.3–100.0)) [5] and has been approved by the Ministry of Health of the Republic of Kenya and the WHO.

To assess mortality, the Dagahaley population was exhaustively sampled. CHWs were asked to visit all households in the camp and to collect data on the total number of household members and every death from the start of the recall period. To calculate age-specific mortality, the estimated age distribution from a population size estimate conducted in 2018 by MSF was used (S1 File). The recall period for the mortality survey was from 1 January 2019 until the start of the study, and was divided into 2 periods: pre-pandemic (prior to COVID-19 detection) from 1 January 2019 to 30 April 2020; and pandemic (presumed active COVID-19 transmission) from 1 May 2020 until the start of the study. Data on the number of consultations, hospitalizations, and deaths at the hospital were extracted from routinely collected MSF data.

## Data analysis

Data analysis was conducted using R (R Core Team, 2020). All indicators (i.e., sex and age) were calculated as proportions with 95% Confidence Intervals (95% CI). Where appropriate, differences in proportions were measured using Pearson $\chi 2$ tests with p-values (p) presented.

Logistic regression was performed and risk-ratios (overall and by subpopulations) calculated. To estimate the risk associated with exposure to another infected person in the household, the first infected person in the household had to be determined and then identify other exposed persons. This was not possible based on symptom onset, given the long recall period, and given the fact that not all participants reported symptoms. Therefore, one of the seropositive persons in the household was randomly selected as the first infected; then this was repeated 1000 times to obtain the relative risk, its confidence interval, and its p-value.

Seroprevalence results were standardized by age group with an age distribution identified in a survey conducted in 2018 by MSF (see S1 File). The proportion of inhabitants was estimated for each five-year age group, then the results were wighted for each individual tested with the inverse of this proportion using the R Survey package (Version 4.0). In addition, adjustement was done taking the estimate, found in the literature, of specificity and sensitivity for identifying previously infected individuals (sensitivity: 95.8% (95% CI: 90.2–100.0), specificity: 98.1% (95% CI: 94.3–100.0)) [5]. A Bayesian estimation of true prevalence was performed from apparent prevalence obtained from individuals tested in the sample using the R Prevalence Package (Version 0.4.0).

The Crude Mortality Rate (CMR) (expressed as deaths/10,000 people/day) was calculated, with 95% Confidence Intervals provided. Mortality was calculated during two periods: 486

days pre-pandemic (1 January 2019 to 31 April 2020) and 326 days from the start of the pandemic (1 May 2020, to 22 March 2021, which corresponds to the median date of the survey implementation).

Use of services was assessed and compared over time based on routinely collected monitoring data. The number of consultations and admissions per individual per year were estimated.

### Patient and public involvement

Although there was no specific public involvement in the development of the protocol, discussions were held with Dagahaley camp community leaders and representatives about the purpose of the study. The network of community health workers in the Dagahaley camp was included in the research team, they were involved in adjusting the questionnaires after the pilot phase and then conducted the retrospective mortality data collection. A work un currently ongoing with the local health authorities on how best to communicate the study results to the community.

### Ethics

This protocol was approved by the MSF Ethics Review Board (V. 1.1, 24.12.2020, ID: 20105) and the KEMRI SERU Scientific and Ethic Review Board (KEMRI/SERU/CIPDR/045/4164). All study procedures involving subject's participation were conducted in compliance with the principles enunciated in the Declaration of Helsinki. Mortality monitoring is part of the routine tasks of CHWs; our activity consisted of strengthening this aspect. Nevertheless, as the mortality survey was part of our study, we sought verbal informed consent from every household, with the designated head of household answering the questionnaire for all relevant members of the household. For the seroprevalence component, written informed consent was sought from all individuals willing to participate. In the case of minors aged between eight to 17 years, participation was proposed for both the minor and the legal guardian. Both the assent of the minor and the consent of the legal guardian were needed for inclusion. In the case of minors below eight years of age, the decision to participate was made solely by the legal guardian. Participant privacy was respected throughout the study.

## Results

### Seroprevalence

In total, 1206 participants were included in the seroprevalence stud,. 662 patients and caretakers were approached, of whom 6.5% (43/662) declined participation. Among the 155 TBAs and CHWs recorded by MSF, 145(94%) participated and were all included. They declared having 818 other family members among whom 442 participated (54%) and were all included. Patients and caretakers represented 51% (619/1206) of the inclusions. The overall proportion of females was 54%. The proportion of participants under 20 years of age was 42% (510/1206), and 17% (204/1206) were 50 years old and above.

The overall proportion with a positive test, either Immunoglobulin G (IgG) or Immunoglobulin M (IgM) positive was 8% (95% CI: 6.6%-9.7%). After adjustment for test specificity and standardization of the seroprevalence results by age group with the age distribution of the camp, the adjusted seroprevalence was estimated at 5.8% (95% CI: 1.6%-8.4%).

The proportion of participants testing positive was lowest among those under 5 years of age and highest among those aged 20 to 34 years and 50 years and above (Table 1).

In multivariate analysis, being 20 years of age and above was associated with an increased risk of seropositivity compared to being less than 20 years old, Relative Risk (RR) = 2.7 (95%

**Table 1. Results of rapid diagnostic test by age group, by sex, and overall, Dagahaley refugee camp, Garisa County, Kenya, May 2021.**

| Age group | Patients and caretakers | | CHW & TBA and their family members | | All | | | | | |
|---|---|---|---|---|---|---|---|---|---|---|
| | Negative | Positive | Negative | Positive | Negative | Positive | 95% CI | IgG + | IgM+ | IgG+ & IgM+ |
| 0–19 Y | 151 (96.8%) | 5 (3.2%) | 333 (94.1%) | 21 (5.9%) | 484 (94.9%) | **26 (5.1%)** | 3.5%-7.4% | 13 (2.5%) | 9 (1.8%) | 4 (0.8%) |
| 0–4 Y | 30 (100%) | 0 (0%) | 83 (97.6%) | 2 (2.4%) | 113 (98.3%) | **2 (1.7%)** | 0.5%-6.1% | 1 (0.9%) | 1 (0.9%) | 0 (0%) |
| 5–9 Y | 43 (97.7%) | 1 (2.3%) | 112 (94.1%) | 7 (5.9%) | 155 (95.1%) | **8 (4.9%)** | 2.5%-9.4% | 6 (3.7%) | 2 (1.2%) | 0 (0%) |
| 10–19 Y | 78 (95.1%) | 4 (4.9%) | 138 (92%) | 12 (8%) | 216 (93.1%) | **16 (6.9%)** | 4.3%-10.9% | 6 (2.6%) | 6 (2.6%) | 4 (1.7%) |
| 20–34 Y | 115 (92.7%) | 9 (7.3%) | 115 (83.9%) | 22 (16.1%) | 230 (88.1%) | **31 (11.9%)** | 8.5%-16.4% | 14 (5.4%) | 10 (3.8%) | 7 (2.7%) |
| 35–49 Y | 166 (94.9%) | 9 (5.1%) | 48 (85.7%) | 8 (14.3%) | 214 (92.6%) | **17 (7.4%)** | 4.6%-11.5% | 10 (4.3%) | 4 (1.7%) | 3 (1.3%) |
| > = 50 Y | 144 (87.8%) | 20 (12.2%) | 37 (92.5%) | 3 (7.5%) | 181 (88.7%) | **23 (11.3%)** | 7.6%-16.3% | 13 (6.4%) | 5 (2.5%) | 5 (2.5%) |
| Female | 214 (92.2%) | 18 (7.8%) | 533 (90.8%) | 26 (9.1%) | 621 (92.4%) | **51 (7.6%)** | 5.8%-9.8% | 24 (3.6%) | 15 (2.2%) | 12 (1.8%) |
| Male | 362 (93.5%) | 25 (6.5%) | 274 (90.7%) | 28 (9.3%) | 488 (91.4%) | **46 (8.6%)** | 6.5%-11.3% | 26 (4.9%) | 13 (2.4%) | 7 (1.3%) |
| **Overall** | 576 (93.1%) | 43 (6.9%) | 533 (90.8%) | 54 (9.2%) | 1109 (92%) | **97 (8%)** | 6.6%-9.7% | 50 (4.1%) | 28 (2.3%) | 19 (1.6%) |

CI = 1.6–4.9, p value = <0.001), while, being female was not associated with significant risk of seropositivity, RR = 0.8 (95% CI = 0.5–1.3, p value = 0.322). Compared to patients and caretakers, being a family member of a CHW or a TBA almost doubled the risk of seropositivity, Relative Risk = 1.9 (95% CI = 1.1–3.2, p value = 0.015), being a CHW or a TBA was also associated with increased risk though this was not statistically significant, RR = 1.5 (95% CI = 0.9–2.6, p value = 0.116).

The recruitment of family members of CHWs and TBAs allowed an exploration of the risk associated with exposure to another seropositive person in the household. Not all households were complete (39% (57/147)), and only households where at least 2 members participated in the survey were included in the analysis. Being exposed to another seropositive person in the household was significantly associated with more than two folds higher risk of seropositivity, RR = 2.7 (95% CI = 1.4–5.2, p value = 0.002). In contrast, the size of the household did not seem to increase the risk, RR = 1.0 (95% CI = 0.9–1.2).

Seropositive participants reported at least one COVID-consistent symptom from January 2020 more frequently compared to seronegative participants (81% versus 45%, p value <0.001) (Table 2). Seropositive participants with at least one COVID-consistent symptom experienced a longer duration of symptoms and a median number of symptoms that was twice as high. In contrast, health seeking behaviour was not statistically different based on seroprevalence status. The symptoms most commonly reported by seropositive participants were cough (70%), headache (64%) and fever (63%).

**Table 2. Characteristics of symptoms associated with COVID-19 and health seeking behaviour by seropositivity and results of the rapid diagnostic test BIOSYNEX COVID-19 BSS, Dagahaley refugee camp, Garisa County, Kenya, May 2021.**

| | Negative | Positive | P value* | IgM + | IgG+ & IgM+ | IgG + |
|---|---|---|---|---|---|---|
| At least one symptom | 45% (500) | 81% (79) | <0.001 | 86% (24) | 84% (16) | 78% (39) |
| Symptoms lasting at least one week | 30% (152) | 52% (41) | 0.016 | 33% (8) | 69% (11) | 56% (22) |
| Median number of symptoms (if at least one) | 3 | 6 | 0 | 5 | 5.5 | 7 |
| Saw a doctor if at least one symptom | 30% (151) | 29% (23) | 1 | 33% (8) | 44% (7) | 21% (8) |
| Went to hospital | 15% (76) | 14% (11) | 1 | 21% (5) | 12% (2) | 10% (4) |
| Hospitalized | 2% (9) | 3% (2) | 0.653 | 4% (1) | 0% (0) | 3% (1) |
| ICU | 0% (2) | 3% (2) | 0.095 | 4% (1) | 0% (0) | 3% (1) |

**Table 3. Mortality rates by age group and period, Dagahaley refugee camp, Garisa County, Kenya.**

| Age group | Pre-Pandemic (1 January 2019-April 2020) | | Pandemic (1 May 2020–22 March 2021) | | | | |
|---|---|---|---|---|---|---|---|
| | Death | Mortality rate | Death | Mortality rate | Rate ratio | 95% CI | p value |
| 0–4 Y | 61 | 0.09 | 58 | 0.12 | 1.30 | 0.91–1.87 | 0.148 |
| 5–19 Y | 10 | 0.01 | 22 | 0.02 | 3.01 | 1.43–6.37 | 0.002 |
| 20–34 Y | 23 | 0.03 | 18 | 0.03 | 1.07 | 0.58–1.99 | 0.824 |
| 35–49 Y | 17 | 0.04 | 18 | 0.06 | 1.45 | 0.75–2.81 | 0.269 |
| > = 50 Y | 89 | 0.35 | 91 | 0.49 | 1.40 | 1.05–1.88 | 0.023 |
| Overall | 200 | 0.05 | 207 | 0.07 | 1.42 | 1.17–1.72 | <0.001 |

## Mortality

During the survey, CHWs recorded the size of all households (n = 12,860) living in Dagahaley and estimated the population of Dagahaley at 88,793 inhabitants. In September 2018, the population was estimated at 72,635 inhabitants; assuming linear growth, the population size mid-pre-pandemic periods was estimated at 78,889 inhabitants and at 85,837 inhabitants mid-pandemic period. In total, 407 deaths were recorded (200 pre-pandemic and 207 during the pandemic).

The CMR was low in both periods (Table 3); however, it increased significantly by 42% during the pandemic. The highest mortality rate was among people aged 50 years and over. The mortality rate increased statistically significantly among 5- to 19-year-olds and those 50 years and over. A sensitivity analysis restricted to the same month of the year for both periods was conducted (pre-pandemic and pandemic) and reached similar results (see S1 File).

In the pre-pandemic period, 44.5% (89/200) of the deceased were aged 50 years and over, while 43.9% (91/207) were aged 50 years and over during the pandemic period. The detailed analysis by month does not show a significant pattern (S1 File).

The main reported cause of death from a known cause was respiratory disease (pre-pandemic = 11%, n = 22, pandemic = 15%, n = 31, p value = 0.35). No significant difference by cause of death was detected during the two periods, even when stratified by aged (<50 vs ≥50 years). COVID-19 was reported as the cause of death for two cases. No significant difference between the pre-pandemic and pandemic periods for any symptoms reported prior to death was detected nor for comorbidities. Most of the deceased had access to the hospital prior to death (pre-pandemic = 74%, (147/199), pandemic = 69% (142/207)). The proportion of deaths in the hospital was similar for both periods (pre-pandemic = 42%, n = 84, pandemic = 36%, n = 75). For more details on the characteristics of the deceased refer to the S1 File.

## Attendance at health facilities

The number of consultations at health centres decreased by more than a third during the pandemic period (Table 4); the decrease started in April 2020 and has yet to return to pre-pandemic levels. At the emergency room, the decrease in consultations was much lower (8%) compared to health centres (38%), with a return to previous levels in October 2020. Admission to the hospital showed a similar decrease as consultations at health centres (see monthly data in S1 File).

## Discussion

The survey found a SARS-COV-2 antibody seroprevalence of 5.8% (95% CI 1.5–8.5) in Dagahaley Refugee Camp. Comparing the pre-pandemic to the pandemic periods, a 42% (95% CI:

**Table 4. Attendance and death at MSF health facilities, Dagahaley refugee camp, Garisa County, Kenya January 2019-March 2021.**

|  | Pre-pandemic (January 2019-April 2020) | | Pandemic (May 2020-March 2021 | | Rate ratio |
|---|---|---|---|---|---|
|  | N | Rate (day) | N | Rate (day) |  |
| Consultation at health centres | 164848 | 339 | 82187 | 245 | 0.72 |
| Consultation at emergency room* | 46105 | 116 | 35961 | 107 | 0.92 |
| Admission to hospital | 12918 | 26.5 | 6467 | 19.3 | 0.73 |
| Death at hospital | 140 | 0.288 | 81 | 0.25 | 0.87 |

*Data from emergency room were available only from April 2019.

17%-72%) increase was observed in the mortality rate as well as a 38% decrease in consultations at health centres, an 8% decrease in emergency room, and a 37% decrease in hospital admissions.

The seroprevalence found in our survey is lower than another survey conducted in Kenya in November 2020 in the Nairobi population which reported a seroprevalence of 34.7% (95% CI 31.8–37.6) [6] based on ELISA, and lower that a national seroprevalence estimated from tests conducted on blood donors [7]. Many factors may have contributed to the lower seroprevalence in our study: first, the isolation of the Dagahaley refugee camp of over 80,000 people and the strict screening of arrivals and visitors in the camp; second, our survey used a rapid test that may not be as sensitive as an ELISA; third, age has been shown to be correlated with transmission [8]—the young age of Dagahaley population could have play a role; fourth, the limited number of meeting places and few large enclosed spaces such as shopping centre in the camp may have limited transmission.

The estimated seroprevalence in this study applied to the population of Dagahaley, comprised of 88,793 inhabitants at the time of the survey, suggesting that 5103 people (95% CI 1332–7455) were infected. This figure is 67 times higher than the number of cases reported in the camp until the last day of the survey (76 confirmed COVID-19 cases up to April 26th 2021).

As the proportion of asymptomatic cases of COVID-19 decreases with age [9], it is possible that the proportion of asymptomatic cases was high in Dagahaley. However, seropositive participants reported having at least one symptom more frequently than seronegative participants (81% versus 45%), with a higher median number of symptoms lasting longer. These results suggest that a substantial proportion of the seropositive were affected by the disease; accordingly, the proportion of asymptomatic cases cannot fully explain the difference between reported cases and estimated infections. Other factors may have been involved, such as the relatively low number of tests performed (less than 1000 for over 80,000 people) and the decrease in access to health care.

The survey showed that the risk of infection was correlated with age. People aged 20–35 years and those aged 50 and over were the most affected in contrast to the Nairobi survey which showed a slightly lower seroprevalence in this age group compared to the 20–50 year olds. Probably due to the intense social mixing of young and old in the camp, the 50+ age group was one of the most exposed, which is worrying as they are at greatest risk of mortality. The survey also shows that having an infected person in the household is another factor that more than doubles the risk of infection; which is consistent with the results of other studies [10].

CHWs, TBAs, and their family members had a higher seroprevalence than patients and caretakers. This suggest that CHWs and TBAs are more exposed to infection, likely by occupational exposure; by extension, their household members are also more exposed due to intra-

household transmission. This calls for improved infection prevention and control measures for this group as well as increased awareness of the importance of early detection of infection to protect the household.

The survey estimated a CMR of 0.05 deaths per day per 10,000 people before the pandemic and 0.07 deaths per day per 10,000 people during the pandemic. These mortality rates are lower than those estimated from reported mortality in Kenya (expected CMR 0.16, see S1 File). This may be due, among other things, to good access to basic health care in the camps. The population has free access to a broad spectrum of primary and secondary care and preventive activities.

Two deaths from COVID-19 were reported during the survey. (Up to the start of the survey, three COVID-19-related deaths were officially reported). Analysis of the causes of death and co-morbidities of the deceased showed no significant differences between the two periods and did not allow the identification of deaths caused by COVID-19 that had not been identified as such. In addition, the proportion of deaths in those over 50 years of age was relatively constant; this suggests no substantial differential increase in mortality in this age group directly from COVID-19.

Eight COVID-19 related deaths would have been expected if the age-specific Infection Fatality Rate estimated by Driscol et al. [11] is applied on the infected population by age group derived from the survey (see detailed calculation in S1 File).

Although one cannot exclude that deaths directly related to COVID-19 were missed, the increase in mortality detected by the survey during the pandemic may also be indirectly caused by decreased access to care as evidenced by decreased attendance at health facilities and other potential adverse effects of the pandemic.

The analysis of data from MSF health facilities showed a significant reduction in attendance during the pandemic compared to the pre-pandemic period. The rates of consultations at primary health centres as well as hospital admissions decreased by a third. However, the number of consultations at the emergency room and the number of hospital deaths decreased less during the period, potentially suggesting that access to health care for severe cases only moderately impacted.

This study has several limitations. For practical reasons, it was not possible to do a population-based sampling to estimate seroprevalence; instead, the families of TBAs and CHWs were targeted as well as patients and caretakers at health centres. This population is potentially different from the camp population, both in terms of exposure to the virus and access to care. However, this population originates from all geographical areas of the camp and includes all ages. The refusal rate was low among patients and caretakers, decreasing the potential risk of selection bias. Among family members of CHW and TBA, none who came to the study site declined to participate, but only 55% participated, which could potentially introduce bias. As the assessment of symptoms was based on participant recall over a one-year period, recall bias may have occurred; however, participants reported symptoms before they knew their test result and a significant difference was detected between seropositive and seronegative individuals.

Due to logistical and time constraints, only rapid tests were used. Rapid tests have less accuracy than laboratory tests such as ELISA. Recent studies have shown differences between results obtained by RDT and ELISA tests, with RDT tending to underestimate seroprevalence [12, 13]. Nevertheless, the BIOSYNEX test showed good performance in studies conducted on subjects infected a few weeks before [5]. Certain types of antibodies decay over time [14], hence the sensitivity of the test may be affected. Cross-reactivity of the test that was used is possible; however, in a study conducted on a pre-pandemic sample from the Central African Republic using three different rapid tests, BIOSYNEX showed the lowest rate of false positivity (1.36% (4/294)) [15].

The increase in mortality may also be due to the long recall period- The number of deaths may have been underreported differentially or misclassified between the pre-pandemic and pandemic periods. Respondents reported to CHWs that 39% of deaths occurred in the hospital (42% in the pre-pandemic period and 36% in the pandemic period). During this period, 221 deaths were recorded in the hospital (140 pre-pandemic deaths and 81 pandemic deaths). Assuming the hospital database is complete and the proportion of in-hospital deaths reported to CHWs is representative, the projection gives 557 deaths (333 pre-pandemic deaths and 224 pandemic deaths), compared to 407 deaths reported to CHWs (200 pre-pandemic and 207 pandemic). This calculation suggests that the survey may have underestimated the mortality rate by 37%; moreover, it suggests that the underestimation was more important in the pre-pandemic period than in the pandemic period (66% versus 3%). Lastly, this increase could be an artefact, as households that would have left the camp during the recall period could not be included.

## Conclusion

This study estimated 67 times more infected cases than the number of cases reported in the camp. This ratio cannot be fully explained by a high proportion of asymptomatic or pauci-symptomatic cases. A significant proportion of the population remains susceptible, and the survey showed that the virus circulates in the camp. The new, more transmissible, and potentially more severe [16, 17] variants could have dramatic consequences in the camp.

The survey showed an increase in the mortality rate during, compared to before, the pandemic. This occurred in tandem with a sustained decline in attendance to health facilities despite the easing of restrictions. Initiatives to restore the population's confidence and access to health care are urgently needed. Furthermore, this study took place in a stable refugee camp with relatively good baseline access to health care as shown by the low CMR. The situation of the more than 80 million forcibly displaced people in the world varies greatly; this merits further research in other sites to assess their situation. Importantly, only large-scale vaccination of the population can limit the risk of excess mortality from COVID-19.

## Supporting information

**S1 Checklist. STROBE statement—checklist of items that should be included in reports of observational studies.**
(DOCX)

**S1 File.**
(DOCX)

## Acknowledgments

We thank the people and community staff of Dagahaley for their participation in the study, the Ministry of Health staff and Médecins Sans Frontières field and headquarters teams for their support and guidance, and Manuel Albela and Rebecca Coulborn for their invaluable advices on analysis and writing.

## Author Contributions

**Conceptualization:** Etienne Gignoux, Frida Athanassiadis, Carole Déglise, Andrew S. Azman, Matilu Mwau, Francisco Luquero.

**Data curation:** Etienne Gignoux, Andrew S. Azman.

**Formal analysis:** Etienne Gignoux, Andrew S. Azman, Francisco Luquero.

**Investigation:** Etienne Gignoux, Ahmed Garat Yarrow, Abdullahi Jimale, Nicole Mubuto, Denis Onsongo Mosoti.

**Methodology:** Etienne Gignoux, Frida Athanassiadis, Ahmed Garat Yarrow, Nicole Mubuto, Carole Déglise, Andrew S. Azman, Matilu Mwau, Francisco Luquero, Iza Ciglenecki.

**Project administration:** Etienne Gignoux, Frida Athanassiadis, Ahmed Garat Yarrow, Abdullahi Jimale, Nicole Mubuto, Carole Déglise.

**Supervision:** Etienne Gignoux, Frida Athanassiadis, Ahmed Garat Yarrow, Nicole Mubuto, Denis Onsongo Mosoti, Matilu Mwau.

**Validation:** Etienne Gignoux, Frida Athanassiadis, Ahmed Garat Yarrow, Andrew S. Azman.

**Visualization:** Etienne Gignoux, Andrew S. Azman.

**Writing – original draft:** Etienne Gignoux, Andrew S. Azman, Matilu Mwau, Francisco Luquero, Iza Ciglenecki.

**Writing – review & editing:** Etienne Gignoux, Frida Athanassiadis, Ahmed Garat Yarrow, Abdullahi Jimale, Nicole Mubuto, Carole Déglise, Andrew S. Azman, Matilu Mwau, Francisco Luquero, Iza Ciglenecki.

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
