## [Decision Letter · Decision Letter 0]

7 Oct 2021

PONE-D-21-26784Seroprevalence of SARS-CoV-2 antibodies and retrospective mortality in a refugee camp, Dagahaley, KenyaPLOS ONE

Dear Dr. Gignoux,

Thank you for submitting your manuscript to PLOS ONE. After careful consideration, we feel that it has merit but does not fully meet PLOS ONE’s publication criteria as it currently stands. Therefore, we invite you to submit a revised version of the manuscript that addresses the points raised during the review process.

We look forward to receiving your revised manuscript.

Kind regards,

Sanjay Kumar Singh Patel, Ph.D.

Academic Editor

PLOS ONE

Journal Requirements:

2. Please include additional information regarding the survey or questionnaire used in the study and ensure that you have provided sufficient details that others could replicate the analyses. For instance, if you developed a questionnaire as part of this study and it is not under a copyright more restrictive than CC-BY, please include a copy, in both the original language and English, as Supporting Information

5. We note you have included a table to which you do not refer in the text of your manuscript. Please ensure that you refer to Table 4 in your text; if accepted, production will need this reference to link the reader to the Table.

Reviewers' comments:

Reviewer's Responses to Questions

**Comments to the Author**

1. Is the manuscript technically sound, and do the data support the conclusions?

Reviewer #1: Yes

Reviewer #2: Partly

2. Has the statistical analysis been performed appropriately and rigorously? 

Reviewer #1: Yes

Reviewer #2: Yes

3. Have the authors made all data underlying the findings in their manuscript fully available?

Reviewer #1: Yes

Reviewer #2: Yes

4. Is the manuscript presented in an intelligible fashion and written in standard English?

Reviewer #1: Yes

Reviewer #2: Yes

5. Review Comments to the Author

Reviewer #1: The manuscript by Gignoux et al. “Seroprevalence of SARS-CoV-2 antibodies and retrospective mortality in a refugee camp, Dagahaley, Kenya” is interesting. This manuscript requires minor revision prior to its publication in PLOS ONE as follows:

Comments

1. The authors should cross-check all abbreviations in the manuscript. Initially, define in full name followed by abbreviation.

2. Lines 55-61, the author should elaborate this paragraph with more information related to Covid-19 symptoms, initial Covid-19 prevention strategy like social distancing and isolation, and role of improving heath immunity via diet or natural-based biomolecules supplementation as anti-Covid-19 (reduction of mortality) with citations – doi: 10.1371/journal.pone.0240878; doi: 10.1007/s12088-020-00908-0; doi: 10.1007/s12088-020-00893-4.

3. The author should avoid the uses of third person i.e., we etc. in the manuscript.

4. Lines 79, how about infectivity and mortality rate of population?

5. Please provide statement about the aim and objectives of this study.

6. The authors may provide at least one Figure as summary or prospect of this study.

Reviewer #2: Please refer to attached file to view all my comments.

---

## [Author Response · Author response to Decision Letter 0]

10 Nov 2021

To the editors

PONE-D-21-26784

Seroprevalence of SARS-CoV-2 antibodies and retrospective mortality in a refugee camp, Dagahaley, Kenya

PLOS ONE

Dear Editors,

Thank you for your time in reviewing our manuscript. We are pleased to submit the revision of our manuscript, “Seroprevalence of SARS-CoV-2 antibodies and retrospective mortality survey in a refugee camp, Dagahaley, Kenya”.

Following your requests please find below our answers or changes we did to the manuscript

1. We have verified that our manuscript meets PLOS ONE's style requirements, including those for file name.

2. We have added the questionnaires used during the survey to the supplementary material

3. The minimal dataset underlying the findings of this study is available on request, in accordance with the legal framework set forth by Médecins Sans Frontières (MSF) data sharing policy (Karunakara U, PLoS Med 2013). MSF is committed to share and disseminate health data from its programs and research in an open, timely, and transparent manner in order to promote health benefits for populations while respecting ethical and legal obligations towards patients, research participants, and their communities. The MSF data sharing policy ensures that data will be available upon request to interested researchers while addressing all security, legal, and ethical concerns. All readers may contact the generic address data.sharing@msf.org or Ms. Aminata Ndiaye (aminata.ndiaye@epicentre.msf.org) to request data.

4. An ethic statement is now included in the methods section.

5. We now refer to table 4 in the text 

6. We have added the caption for supplementary information file at the end of the manuscript. 

7. We have revised the references list.

Please do not hesitate to contact me with any questions. We look forward to hearing from you after you have had time to review our manuscript. Many thanks for your time and consideration.

Sincerely, 

Etienne GIGNOUX, MPH MSc

Epicentre

78 rue de Lausanne

1202 Geneva

Switzerland

Tel : +41 228498231

E-mail : etienne.gignoux@geneva.msf.org

To the reviewers:

Reviewer #1: The manuscript by Gignoux et al. “Seroprevalence of SARS-CoV-2 antibodies and retrospective mortality in a refugee camp, Dagahaley, Kenya” is interesting. This manuscript requires minor revision prior to its publication in PLOS ONE as follows:

Comments

1. The authors should cross-check all abbreviations in the manuscript. Initially, define in full name followed by abbreviation.

Response:

Thank you for this comment, we added the following abbreviations in the text (following the full name) and in the list of abbreviations.

CI: Confidence Intervals

ELISA : Enzyme-linked Immunosorbent Assay

IgG: Immunoglobulin G

IgM: Immunoglobulin M

PCR : Polymerase Chain Reaction

RR: Relative Risk

WHO : World Health Organization

2. Lines 55-61, the author should elaborate this paragraph with more information related to Covid-19 symptoms, initial Covid-19 prevention strategy like social distancing and isolation, and role of improving heath immunity via diet or natural-based biomolecules supplementation as anti-Covid-19 (reduction of mortality) with citations – doi: 10.1371/journal.pone.0240878; doi: 10.1007/s12088-020-00908-0; doi: 10.1007/s12088-020-00893-4

Response:

The purpose of the first two paragraphs of the background section is to situate the problem: the initial models predicted a high attack rate in camp of forcibly displaced population, but the actual number of cases reported were far below these estimates. The question addressed in this article is whether this number of cases reflects reality or whether it is a reporting problem. 

We have presented the causes explained by the researchers who made the initial predictions. It is possible, however, that we did not elaborate enough in the second paragraph on the reasons for the low number of reported cases. We thank the reviewer for the article suggestions, as the social network model may explain more complex dynamics than the SIR models. However, we prefer to avoid discussing prevention and treatment methods that are still in the study stages as we think it is out of the scope of this study. We recognize, however, that unknown factors associated with these populations may be the reason for the low number of reported cases. Therefore we have rephrased as (lines 65-69 of the revised manuscript):

“The low numbers have been attributed to limited testing capacity, differences in the population structure with a small proportion of elderly at high risk of severe disease and death, a predominance of asymptomatic and pauci-symptomatic infections, early implementation of confinement(3), social structure leading to different epidemic dynamics(4), or other unknown factors associated with this population and the context”

3. The author should avoid the uses of third person i.e., we etc. in the manuscript.

Response:

Thank you for this comment, we have corrected the manuscript accordingly

4. Lines 79, how about infectivity and mortality rate of population?

Thank you for the suggestion. We rephrased as (line 86-88 of the revised manuscript)::

“As of the start of the study on 3 March 2021, 106,470 confirmed cases of COVID-19 had been reported in Kenya including 1,863 deaths (source World Health Organization -WHO). Based on the 2019 national census, this corresponds to an attack rate of 0.22% and 3.9 COVID-19 related deaths per 100,000 population.”

5. Please provide statement about the aim and objectives of this study.

to be clearer we rephrased the text as (line 68-71 of the revised manuscript):

“Therefore, the aim of the study was to provide a more accurate picture of the extent of the epidemic, the specific objectives were to estimate the seroprevalence of SARS-COV-2 through a cross-sectional survey; in addition, through a retrospective survey and programmatic data, to assess its impact on mortality and access to care before and during the COVID-19 epidemic.”

6. The authors may provide at least one Figure as summary or prospect of this study.

The main messages of our study are a) higher than expected seroprevalence, b) higher mortality rate during the pandemic, c) reduced access to care during the pandemic.

Your suggestion is interesting, nevertheless we are not sure to have understood it clearly: do you mean figure as a graph, or as a number? . If you suggest a graph, for point a), a table is more appropriate. For point b) and c) we could have used a graph, but as the pre-pandemic period and the pandemic period are not of the same duration, the reading is more difficult and we preferred to represent rates in tables. However, the graphs per month of deaths and access to care are in the supplementary material. If you mean a number, we could make a box in which we would present the main results a,b,c.

Reviewer #2: 

 Summary: 

The authors of this paper investigated the seroprevalence of SARS-CoV-2 antibodies and estimated the mortality in a refugee camp located in Dadaab refugee complex, Kenya. Rapid immunologic assay (BIOSYNEX® COVID-19 BSS [IgG/IgM] were used to determine seroprevalence. The authors interviewed all households (n=12860) in the camp to report the mortality rate. The study identified age groups that were most affected and observed an increase in the mortality rate during the pandemic. 

Major comments: 

1. The authors said “During the survey, CHWs recorded the size of all households (n=12,860) living in Dagahaley and estimated the population of Dagahaley at 88,793 inhabitants.” They also reports “ In total, we included 1206 participants in the seroprevalence study”. So, this means they have sampled 1.35 % of the entire population, therefore this study is not broad enough. To make more informed decision the authors should increase the sampling %. 

Response:

In the protocol the sample was designed to give a precision of plus or minus 5% per age group for crude seroprevalence which was planned to give for all age a precision of plus or minus 3%, which is what we obtained :”The overall proportion with a positive test, …was 8% (95% CI: 6.6%-9.7%)”. This calculation is independent of the population size as the size of the sample is small compare to the population size ((population size-sample size)/(population size-1) ≈1) (https://journals.sagepub.com/doi/full/10.1177/104063870902100102). In addition, to reassure the reviewer that our sample size is reasonable, the survey of the entire population of Nairobi included 1164 individuals, like ours.

However, as we explained in the methods, we were not able to conduct random sampling, and we describe how we selected the sample: “The study entailed a cross-sectional survey using convenience sampling to estimate seroprevalence. A population-based survey was not deemed feasible in this setting due to limited access by MSF staff to the camp and expectedly low participation rates in the camp. To estimate seroprevalence, two cohorts accessible to trained MSF staff were included: a) All CHWs and TBAs working in Dagahaley camp and their household members, as well as b) Patients and caretakers presenting at MSF-supported health centres in Dagahaley camp. “. The limitations linked to convenience sampling are discussed in discussion section.

2. It is quite surprising that the seroprevalence found this survey is significantly lower than Nairobi population which reported a seroprevalence of 34.7%. I am wondering if the authors have cross verified the rapid antibody test negative subjects with an ELISA test. 

Response:

The difference in seroprevalence between Nairobi and Dagahaley may seem surprising, but not necessarily unexpected. The distance as the crow flies between the two locations is over 400 km. More importantly, the contexts are different: Dagahaley is a remote refugee camp in the middle of the "desert," while Nairobi is the city's capital. We do not have a clear understanding of the factors that explain the high seroprevalence in Nairobi, however in the discussion we present our understanding of the lower prevalence in Dagahaley where our study took place: “Many factors may have contributed to the lower seroprevalence in our study: first, the isolation of the Dagahaley refugee camp of over 80,000 people and the strict screening of arrivals and visitors in the camp; second, our survey used a rapid test that may not be as sensitive as an ELISA; third, age has been shown to be correlated with transmission (8)—the young age of Dagahaley population could have play a role; fourth, the limited number of meeting places and few large enclosed spaces such as shopping centre in the camp may have limited transmission.”

Only rapid tests were used due to logistical and time constraints. We agree with the reviewer that the use of an ELISA test could have brought more precision in our estimation. We may have not highlight enough that the use of RDTs may have underestimated seroprevalence; Therefore we have rephrased the discussion (Line 325-329 of the revised manuscript) :

“Due to logistical and time constraints, only rapid tests were used. Rapid tests have less accuracy than laboratory tests such as ELISA. Recent studies have shown differences between results obtained by RDT and ELISA tests, with RDT tending to underestimate seroprevalence(12)(13). Nevertheless, the BIOSYNEX test showed good performance in studies conducted on subjects infected a few weeks before (5)….”

Minor comments: 

1. Line 48: change where to were. 

Thank you for noticing that. This is now corrected

---

## [Decision Letter · Decision Letter 1]

22 Nov 2021

Seroprevalence of SARS-CoV-2 antibodies and retrospective mortality in a refugee camp, Dagahaley, Kenya

PONE-D-21-26784R1

Dear Dr. Gignoux,

We’re pleased to inform you that your manuscript has been judged scientifically suitable for publication and will be formally accepted for publication once it meets all outstanding technical requirements.

Kind regards,

Sanjay Kumar Singh Patel, Ph.D.

Academic Editor

PLOS ONE

---

## [Editor Report · Acceptance letter]

9 Dec 2021

PONE-D-21-26784R1 

Seroprevalence of SARS-CoV-2 antibodies and retrospective mortality in a refugee camp, Dagahaley, Kenya 

Dear Dr. Gignoux:

I'm pleased to inform you that your manuscript has been deemed suitable for publication in PLOS ONE. Congratulations! Your manuscript is now with our production department. 

Kind regards, 

on behalf of

Dr. Sanjay Kumar Singh Patel 

Academic Editor

PLOS ONE